# Copper Oxide Films Deposited by Microwave Assisted Alkaline Chemical Bath

Reina Galeazzi Isasmendi [1],[*], Isidro Juvenal Gonzalez Panzo [1], Crisóforo Morales-Ruiz [1], Román Romano Trujillo [1], Enrique Rosendo [1], Iván García [1], Antonio Coyopol [1], Godofredo García-Salgado [1], Rutilo Silva-González [2], Iván Oliva Arias [3] and Carolina Tabasco Novelo [4]

1 CIDS-ICUAP, Benemérita Universidad Autónoma de Puebla, 14 Sur y Av. San Claudio s/n, C.U. Col San Manuel, Edif. IC-6, Puebla Pue 72570, Mexico; isjuv.gp@gmail.com (I.J.G.P.); crisomr@yahoo.com.mx (C.M.-R.); roman.romano@gmail.com (R.R.T.); enrique171204@gmail.com (E.R.); iegarcia@hotmail.com (I.G.); acoyopol@gmail.com (A.C.); godgarcia@yahoo.com (G.G.-S.)
2 Instituto de Física, Benemérita Universidad Autónoma de Puebla, Apdo. Postal J-48, Puebla Pue 72570, Mexico; silva@ifuap.buap.mx
3 Centro de Investigación y de Estudios Avanzados del IPN Unidad Mérida, Departamento de Física Aplicada, A.P. 73-Cordemex, Mérida 97310, Mexico; andresivanolivaarias@gmail.com
4 Departamento de Física y Matemáticas, Instituto de Ingeniería y Tecnología, Universidad Autónoma de Ciudad Juárez, Av. Del Charro 450 Col. Romero Partido, C. P., Juárez 32310, Mexico; carolinatabasco@gmail.com
* Correspondence: ingquim25@gmail.com; Tel.: +52-(222)-2295500 (ext. 7876)

**Abstract:** Copper oxide (CuO) films were deposited onto glass substrates by the microwave assisted chemical bath deposition method, and varying the pH of the solution. The pH range was varied from 11.0 to 13.5, and the effects on the film properties were studied. An analytical study of the precursor solution was proposed to describe and understand the chemical reaction mechanisms that take place in the chemical bath at certain pH to produce the CuO film. A series of experiments were performed by varying the parameters of the analytical model from which the CuO films were obtained. The crystalline structure of the CuO films was studied using X-ray diffraction, while the surface morphology, chemical composition, and optical band-gap energy were analyzed by scanning electron microscopy, X-ray photoelectron spectroscopy, and UV–Vis spectrophotometry, respectively. The CuO films obtained exhibited a monoclinic crystalline phase, nanostructured surface morphology, stoichiometric Cu/O ratio of 50/50 at%, and band-gap energy value of 1.2 eV.

**Keywords:** semiconductors; precipitation; growth processes; nanostructures; characterization

## 1. Introduction

Metal oxide nanostructured materials are of particular interest for the development of novel functional and smart materials. Copper oxide (CuO) is studied for its optical properties and applications as a p-type semiconductor, with a direct bandgap energy of 1.2 eV. CuO has been investigated for electronic, magnetic, and optical applications, since it can absorb light up to the near infrared region [1–4]. In addition, CuO nanostructures with various morphologies and sizes have been obtained by different methods of synthesis [5]. The chemical bath deposition (CBD) technique has a low cost, simplicity, reproducibility, and suitability for large-scale production [6]. However, the CBD growing mechanisms responsible for the CuO nanostructure formation have not been clearly elucidated; although the chemical reagent effects have been studied. S.K. Shinde et al. [7] reported the effect of different ionic liquid solvents on the properties of CuO thin films; the chemical precursors in the bath were copper sulfate as the copper source and aqueous ammonia as the complexing agent, while monoclinic phase of CuO and nanosheet-like and hybrid nanostructures of the surface morphology were obtained on the CuO films for supercapacitor applications. On the other hand, S. Valanarasu et al. [6] used copper chloride and aqueous ammonia in the

chemical bath. The structural, optical, and electrical properties of the synthesized CuO films were reported as a function of the pH solution (10–11 range). J. Sultana et al. [8] reported an interesting growth kinetic of CuO thin films; copper chloride and aqueous ammonia were used in the bath. Aqueous ammonia (NH$_4$OH) plays an important role because it provides the NH$_4^+$ and OH$^-$ ions that are required for CuO growth in the solution. CuO film deposition by CBD needs a metal salt as a copper source and a complexing agent; copper nitrate, sulfate, chloride, and acetate are frequently used as a metal source, while aqueous ammonia, sodium hydroxide, and ammonium persulfate are used as complexing agents [9–11]. In the literature, a chemical route for depositing the CuO film is explained; however, the CBD growing mechanism is not completely clear and only is explained through the aqueous intermediate Cu(OH)$_2$ species. Therefore, an accurate detection of precursors, the development of morphology formation, and development of analytical models for explaining the growing mechanism processes are required.

The CBD CuO film precipitation process is still not clear, and there is not enough information about the effects of the reaction conditions on the properties of nano-sized CuO; for this reason, a greater knowledge about the synthesis of CuO is required to understand how the reactions in alkaline aqueous medium work and to allow the formation of CuO with different sizes, shapes, and chemical ratios, when the reaction conditions of the solution are varied.

In this work, the effect of the solution pH on the reaction products was studied as a function of the formed chemical species, which appear during the chemical bath deposition as a result of the used chemical reagent concentrations. For this purpose, an analytical model predicting these species was proposed by considering the main chemical reactions that take place in the chemical bath. Moreover, a comparison of the obtained results, between the thermodynamic model and the experimental technique, allowed us to discuss the relation of the theoretical results as a function of different pH values and their influence over the physical and chemical characteristics of the obtained CuO nanostructures. On the other hand, the use of microwave irradiation for nanoparticle preparation has also been reported [12]. Microwave irradiation has shown a rapid growth for material science applications, due to its unique reaction effects such as rapid volumetric heating and the acute increase in the reaction speed; in consequence, microwave synthesis has the advantages of short reaction times, a small particle size, narrow particle size distribution, and high purity in comparison with conventional methods [13].

The aim of this work was to investigate the influence of the starting reaction conditions on the physical and chemical properties of CuO nanostructures synthesized by microwave chemical bath deposition (MWCBD) when the pH in the aqueous solution is varied. The results of the proposed analytical model were compared with the experimental data. The strategy adopted for obtaining the CuO nano or microstructures was to use the predictions described by the species distribution diagrams (SDDs) obtained under certain chemical conditions. The predicted species present in the aqueous solution as a function of the pH, define their influence on the physical and chemical properties of the nanostructures obtained. Furthermore, these intermediate aqueous species elucidate the growing mechanism processes and give a better understanding of CuO film preparation by the chemical bath deposition method and, therefore, of how to obtain desirable physical and chemical properties in the film.

## 2. Physical-Chemical Analysis

*Solubility and Species Distribution Diagrams*

According to the chemical reagents used in this work: copper acetate (Cu(CH$_3$COO)$_2$), ammonium hydroxide (NH$_4$OH), and urea (CH$_4$N$_2$O) (the latter mainly used to control the reaction due to its chelating effect [14]), the possible chemical reactions in the solution are:

Ammonium dissociation [15,16]:

$$\text{NH}_4^+{}_{(aq)} \leftrightarrow NH_{3(aq)} + H^+{}_{(aq)} \quad \text{Log K}_1 = 9.244 \ 25°, \ 0 \tag{1}$$

Copper-amino complexes formation [16,17]:

$$Cu^{2+}{}_{(aq)} + NH_{3(aq)} \leftrightarrow Cu(NH_3)^{2+}{}_{(aq)} \ \text{Log K}_2 = 4.04 \ 25°, 0 \tag{2}$$

$$Cu^{2+}{}_{(aq)} + 2NH_{3(aq)} \leftrightarrow Cu(NH_3)_2^{2+}{}_{(aq)} \ \text{Log K}_3 = 7.47 \ 25°, 0 \tag{3}$$

$$Cu^{2+}{}_{(aq)} + 3NH_{3(aq)} \leftrightarrow Cu(NH_3)_3^{2+}{}_{(aq)} \ \text{Log K}_4 = 10.27 \ 25°, 0 \tag{4}$$

$$Cu^{2+}{}_{(aq)} + 4NH_{3(aq)} \leftrightarrow Cu(NH_3)_4^{2+}{}_{(aq)} \ \text{Log K}_5 = 11.75 \ 25°, 0 \tag{5}$$

Copper-hydroxy complexes formation [17,18]:

$$Cu^{2+}{}_{(aq)} + OH^-{}_{(aq)} \leftrightarrow Cu(OH)^+{}_{(aq)} \ \text{Log K}_6 = 6.3 \ 25°, 0 \tag{6}$$

$$Cu^{2+}{}_{(aq)} + 2OH^-{}_{(aq)} \leftrightarrow Cu(OH)_2^0{}_{(aq)} \ \text{Log K}_7 = 12.8 \ 25°, 0 \tag{7}$$

$$Cu^{2+}{}_{(aq)} + 3OH^-{}_{(aq)} \leftrightarrow Cu(OH)_3^-{}_{(aq)} \ \text{Log K}_8 = 14.5 \ 25°, 0 \tag{8}$$

$$Cu^{2+}{}_{(aq)} + 4OH^-{}_{(aq)} \leftrightarrow Cu(OH)_4^{2-}{}_{(aq)} \ \text{Log K}_9 = 16.4 \ 25°, 0 \tag{9}$$

Water dissociation [19]:

$$H_2O_{(aq)} \leftrightarrow H^+{}_{(aq)} + OH^-{}_{(aq)} \ \text{Log K}_{10} = 13.997 \ 25°, 0 \tag{10}$$

Precipitated copper formation [19]:

$$Cu(OH)_{2(s)} \leftrightarrow Cu^{2+}{}_{(aq)} + 2OH^-{}_{(aq)} \ \text{Log K}_{11} = 19.66 \ 25°, 0 \tag{11}$$

This work takes advantage of previous research [20,21], and similar calculations were made to develop the analytical model, providing the solubility and species distribution diagrams (SDs and SDDs) that help us to predict the chemical conditions for the deposition of the CuO films by the microwave activated chemical bath method (MWCBD). The stability constant of the equations from (1) to (11) is given as 25 °C and infinite dilution (25°, 0). The mass balance of the used reactant and the charge balance of the solution are as follows:

Ammonium mass balance:

$$[NH_4^+] = [NH_3] + [NH_4^+] + \left[Cu(NH_3)^{2+}\right] + 2\left[Cu(NH_3)_2^{2+}\right] + 3\left[Cu(NH_3)_3^{2+}\right] + 4\left[Cu(NH_3)_4^{2+}\right] \tag{12}$$

Copper mass balance:

$$[Cu^{2+}] = [Cu(OH)_2]_{(s)} + [Cu^{2+}] + \left[Cu(OH)^+\right] + \left[Cu(OH)_2^0\right] + \left[Cu(OH)_3^-\right] + \left[Cu(OH)_4^{2-}\right] +$$
$$\left[Cu(NH_3)^{2+}\right] + \left[Cu(NH_3)_2^{2+}\right] + \left[Cu(NH_3)_3^{2+}\right] + \left[Cu(NH_3)_4^{2+}\right] \tag{13}$$

Solution charge balance:

$$Q_P^+ = [H^+] + \left[Cu(OH)^+\right] + [NH_4^+] + 2[Cu^{2+}]$$
$$+2\left[Cu(NH_3)^{2+}\right] + 2\left[Cu(NH_3)_2^{2+}\right] + 2\left[Cu(NH_3)_3^{2+}\right] + 2\left[Cu(NH_3)_4^{2+}\right]$$
$$Q_P^- = [OH^-] + \left[Cu(OH)_3^-\right] + \left[(CH_3COO)^-\right] + 2\left[Cu(OH)_4^{2-}\right]$$
$$Q_P^+ - Q_P^- = 0 \tag{14}$$

Now, an analytical model predicting the maximum copper acetate solubility is determined by the chemical equilibrium (Equations (1)–(11)) and by the mass and charge balance (Equations (12)–(14)). For simplicity, the coefficient activities of all chemical species are neglected and [H⁺] is varied indirectly within the interval from $10^{-3}$ to $10^{-14}$ by means of the $[NH_4OH]_{total}$ to obtain an analytical model as a function of pH. As usual, [OH⁻] and [H⁺]

are related by the water dissociation. Then, $[Cu^{2+}]$ can be estimated from Equation (11), since the precipitate $Cu(OH)_2$ must be formed, as follows:

$$\left[Cu^{2+}\right] = \frac{K_{11}}{[OH^-]^2} \tag{15}$$

The $[NH_3]$ concentration in the solution can be determined by the ammonium mass balance and the chemical equilibrium expressions (Equations (2)–(5)):

$$\left[Cu^{2+}\right] \sum_{i=2}^{5}\left(K_i[NH_3]^{i-1}\right) + [NH_3]\left(1 + \frac{[H^+]}{K_1}\right) - [NH_4OH]_{total} = 0 \tag{16}$$

and finally the $[Cu(CH_3\,COO)_2]_{total}$ is calculated with Equation (13):

$$[Cu(CH_3COO)_2]_{total} = \left[Cu^{2+}\right]\left(1 + \sum_{i=2}^{5} K_i[NH_3]^{i-1} + \sum_{i=6}^{9} K_i\left[OH^-\right]^{i-5}\right) \tag{17}$$

For the construction of SDDs, the chelate $[Cu(urea)]^{2+}$ formation is not included, because its main function is the slow release of $Cu^{2+}$ ions into the solution, in order to obtain a controlled MWCBD reaction. On the other hand, urea thermally decomposes in aqueous media. The urea decomposition reaction consists in a series of steps, where the time plays an important role. The film deposition by the MWCBD technique takes a short period of time in contrast with the urea decomposition reaction. This ensures that the aqueous solution is not contaminated with the products of the urea during the film deposition.

The species distribution diagrams (SDDs) were constructed by varying the ammonium hydroxide ($NH_4OH$) content in order to obtain different pH values in the chemical bath solution. Figure 1 shows the SDDs at 25 °C for pH values from 11 to 13.5. Different chemical species can be seen: $Cu^{2+}$, $Cu(NH_3)^{2+}$, $Cu(NH_3)_2^{2+}$, $Cu(NH_3)_3^{2+}$, $Cu(NH_3)_4^{2+}$, $Cu(OH)^+$, $Cu(OH)_2^0$, $Cu(OH)_3^-$, and $Cu(OH)_4^{2-}$. The presence of these species affects the nucleation and the growth rate during material deposition. Each vertical dotted line corresponds to a different pH value. At pH = 11, the $Cu(OH)_2^0$ species dominate. At pH = 11.5, the $Cu(OH)_3^-$ species begins to increase. Moreover, for pH = 12 and pH = 12.5, the $Cu(OH)_2^0$, $Cu(OH)_3^-$, and $Cu(OH)_4^{2-}$ species are in higher proportion. The $Cu(OH)_2^0$ dominate at pH = 12 and the $Cu(OH)_4^{2-}$ at pH = 12.5, while the $Cu(OH)_3^-$ remains constant. For high pH values of 13 and 13.5 the $Cu(OH)_4^{2-}$ species dominate. The copper-amino complexes $Cu(NH_3)^{2+}$, $Cu(NH_3)_2^{2+}$, $Cu(NH_3)_3^{2+}$, and $Cu(NH_3)_4^{2+}$ increase with pH; however, these last pH values were out of range in our study.

On the other hand, the copper-hydroxy complexes species, $Cu(OH)_2^0$, $Cu(OH)_3^-$, and $Cu(OH)_4^{2-}$ are the main products in the solution for CuO formation. For example, the $Cu(OH)_2^0$ species must initiate the growth (nuclei), while the cuprates ions $Cu(OH)_3^-$ and $Cu(OH)_4^{2-}$ are useful during growth. Similar considerations were found in a previous work [21], the species with zero charge $Cu(OH)_2^0$ are responsible for CuO nucleation, while the species with negative charge $Cu(OH)_3^-$ and $Cu(OH)_4^{2-}$ are responsible for CuO growth, this statement will be verified with the experimental results.

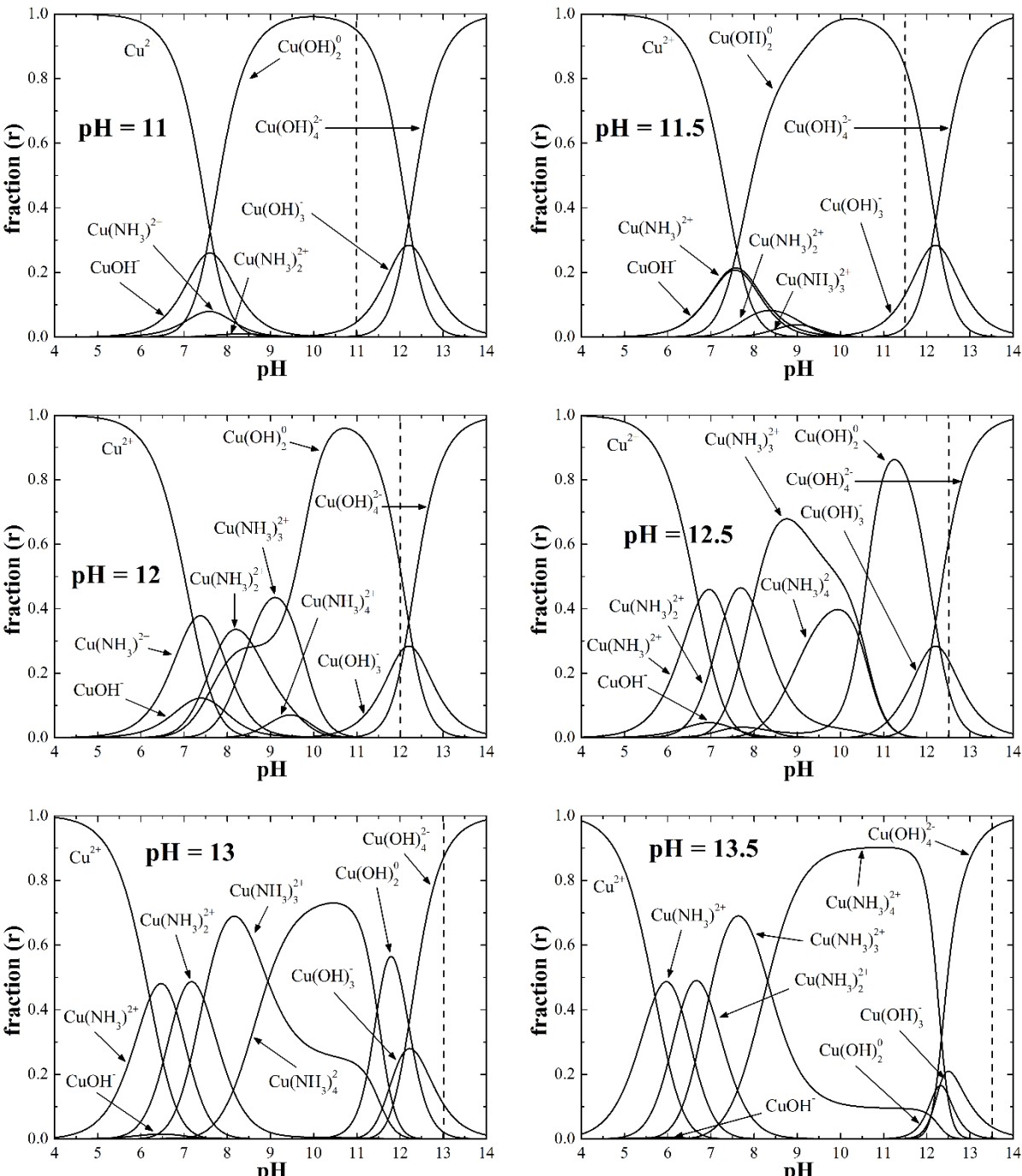

**Figure 1.** Species distribution diagrams (SDDs) obtained at 25 °C for pH values from 11 to 13.5. The pH value is fixed by varying the ammonium hydroxide ($NH_4OH$) concentration.

Figure 2 shows the solubility diagrams (SDs) at 25 °C for the different pH values fixed by varying the ammonium hydroxide ($NH_4OH$) concentration. The used $NH_4OH$ concentration is plotted with different color line and the respective pH value in solution is indicated by a square symbol with the same color. At pH = 11, the minimum solubility in aqueous solution was obtained. Thermodynamically, at this value of pH, the CuO formation is more favorable and the solubility increases, but that does not mean that CuO formation occurs. On the other hand, a large change in solubility of about $10^{-6}$ to $10^{-3}$ can be observed in the SDs as the pH increases. It would be interesting to know how the physicochemical conditions given by the SDDs and SDs can influence the CuO film growth

and the film properties, as well as the formed chemical species and the solubility of each $NH_4OH$ concentration.

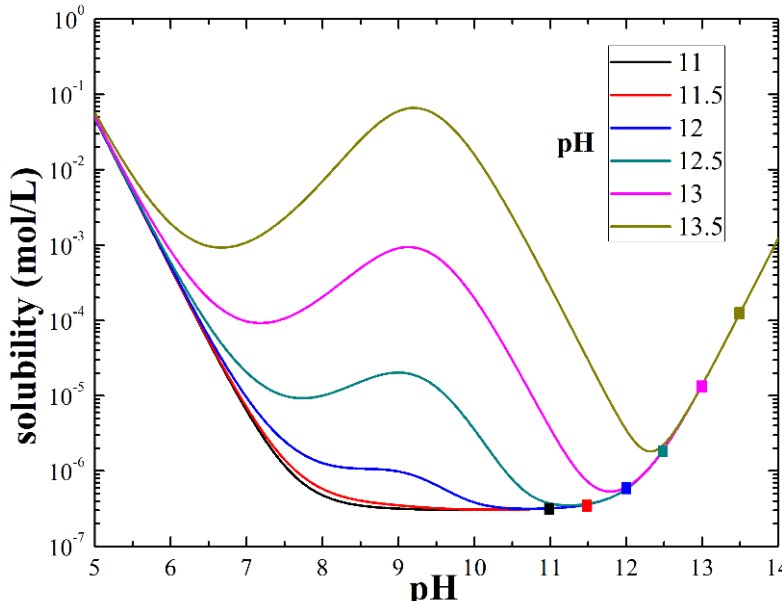

**Figure 2.** Solubility diagrams obtained at 25 °C for each pH value fixed by the ammonium hydroxide ($NH_4OH$) concentration.

## 3. Experimental Procedure

The CuO nano and microstructures were synthesized by MWCBD using copper acetate ($Cu(CH_3COO)_2$) (0.1 M), urea ($CH_4N_2O$) 0.1 M, and different quantities of ammonium hydroxide ($NH_4OH$) to obtain the different pH values in the bath (11.0 to 13.5 with steps of 0.5). In order to investigate the influence of the reaction products described by the SDDs, an initial solution was prepared at room temperature by dissolving zinc nitrate (0.1 M) and urea (0.1 M) in deionized water (DW) (electrical resistivity 18.2 M$\Omega$) under magnetic stirring. Different amounts of aqueous ammonia ($NH_4OH$) were added into the solutions in order to adjust the initial concentrations of $OH^-$ ions ([$OH^-$], mol/L) to: (a) $10^{-3}$, (b) $10^{-2.5}$, (c) $10^{-2}$, (d) $10^{-1.5}$, (e) $10^{-1}$, and (f) $10^{-0.5}$. These values were estimated by measuring the pH value in each solution with a pH-meter at 25 °C. Three well-cleaned glass substrates were vertically immersed into the solution, introduced into a commercial microwave oven (Sharp, model R-408J, 2.45 GHz), and irradiated for 5 min at full power (1200 W) until the aqueous solution reached 80 $\pm$ 2 °C. Finally, the substrates were taken out of the reaction beaker, rinsed several times with DW under vigorous stirring, and dried at room temperature in air.

For obtaining the deposited CuO film properties, the X-ray diffraction (XRD) technique was used to investigate the crystalline structure. Morphological characterization was carried out using a JEOL (JSM-5400LV) scanning electron microscope (SEM). For optical analysis and bandgap energy value determination, an UV-Vis spectrophotometer (AE-UV1608PC) was used to obtain the absorbance profiles of the samples in the 500–1100 nm wavelength range. For chemical analysis, X-ray photoelectron spectroscopy (XPS) measurements were performed using a spectrometer (Thermo Scientific) with an Al-K$\alpha$ source.

## 4. Results and Discussion

### 4.1. X-ray Analysis

The influence of the pH on the structure and properties of the CuO films was investigated. Figure 3 shows the XRD patterns of the as-prepared CuO films with the MWCBD method for pH values from 11 to 13.5. Different colors in the plot signify different pH values in Figure 3. All diffraction peaks in Figure 3 are in good concordance with the

standard patterns of the monoclinic phase of CuO (JCPDS No. 80-1268) [22], and no additional peaks of other phases or impurities were detected, except for films prepared at pH = 11, where no diffraction peaks were detected, due to the low crystallinity of the film. The diffraction peaks of the as-prepared samples were well defined, revealing the high crystallinity of the CuO samples. Moreover, for samples at pH = 12, well defined and high intensity diffraction peaks can be observed. On the other hand, the monoclinic phase of CuO has versatile characteristics that can be used as a dielectric [23] and in batteries [22], as well as having good optical and catalytic properties [24], etc. Therefore, the as-prepared CuO films in this work have an appropriate structure for use in different applications.

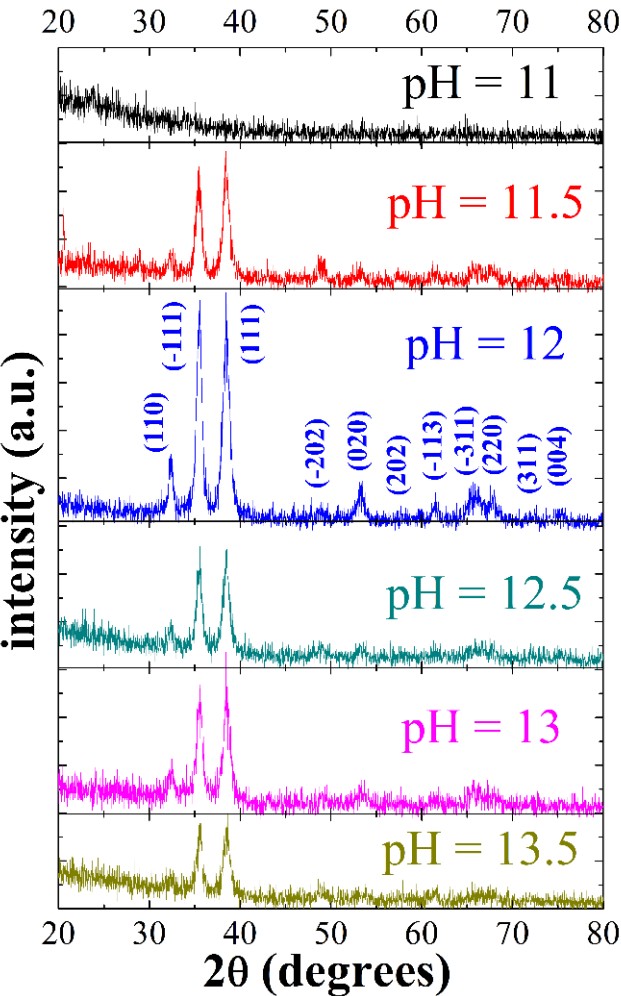

**Figure 3.** X-ray diffraction patterns of the as-prepared MWCBD-CuO films with different pH values.

The influence of secondary chemical reactions due to the solution solubility, and the $Cu(OH)_2^0$, $Cu(OH)_3^-$, and $Cu(OH)_4^{2-}$ species, can be observed for pH values from 11.5 to 13.5 (see Figures 1 and 2), which is a notably a lower intensity than the (110), (−111), and (111) diffraction peaks in Figure 3. The copper-hydroxy complexes, their relative fraction (r), and solution solubility with pH affect the nucleation and the growing rate of the CuO film in specific crystallographic orientations.

### 4.2. Morphology Analysis (SEM)

The 2D surface morphology of the MWCBD-CuO films obtained at different pH values was revealed by using the scanning electron microscopy (SEM) technique and is shown in Figure 4, where different morphologies can be seen. For films at pH = 11, stacked flattened bars are observed. On the right, a close-up of the image permits observing this morphology and its nanostructure nature in detail. For films at pH = 11.5, clusters of pointed end sheets

are observed, and some of them are stacked in a spherical geometry. For samples with pH values of 12, 12.5, 13, and 13.5 a similar morphology is observed; pointed end sheets and vertical clusters aligned along the *z*-axis. Moreover, a notable difference in the size of the sheets and their alignment is observed. In Figure 4, note that the MWCBD-CuO films show a characteristic pointed end sheet morphology, except for the CuO films obtained at pH = 11.

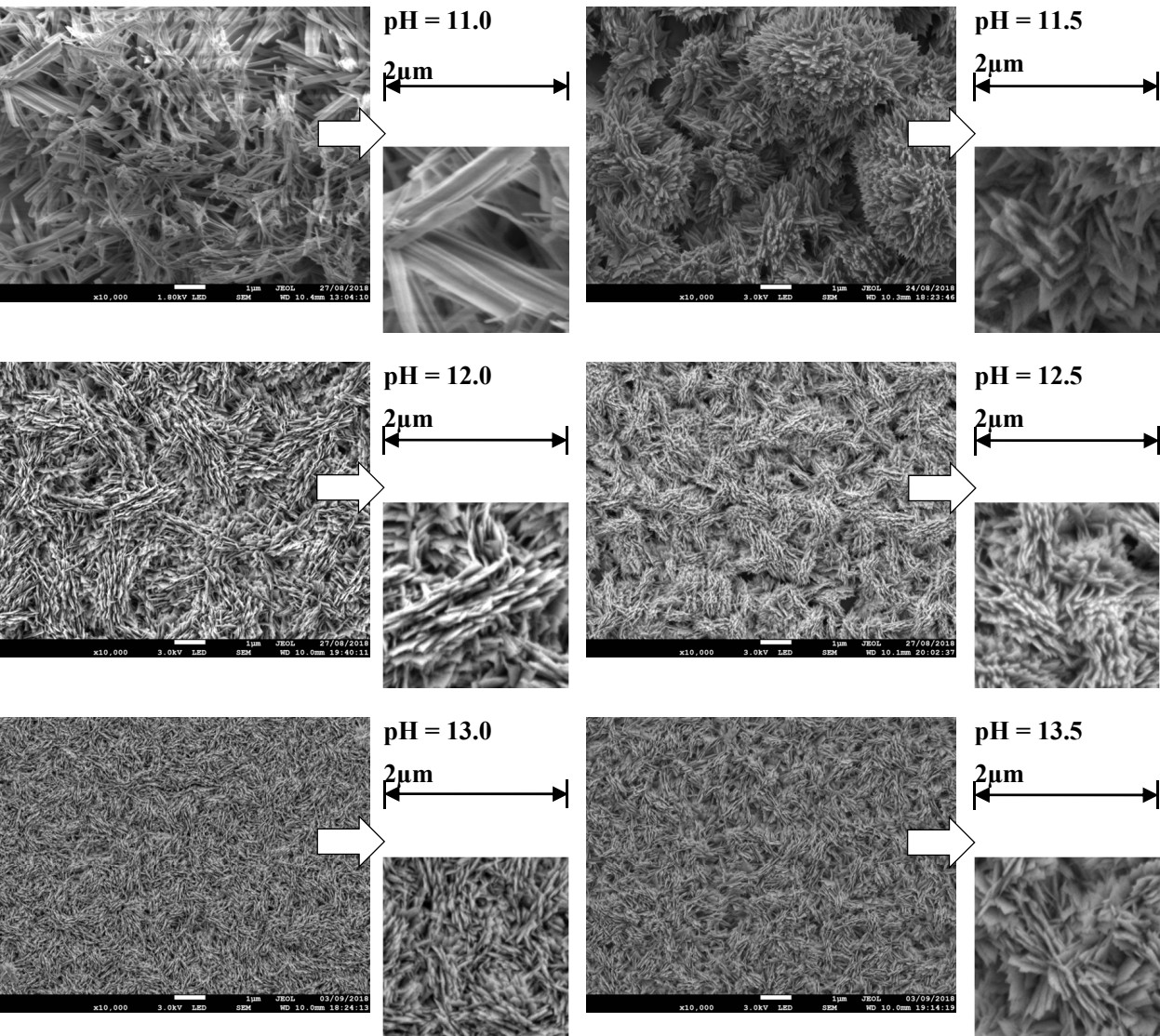

**Figure 4.** SEM images of the as-prepared MWCBD-CuO films at different pH values.

The 11 to 13.5 range of pH with steps of 0.5 in the aqueous solution modified the surface morphology of the CuO films, this could have originated from the solubility changes and the influence of the zinc-hydroxy complexes. For example, the $Cu(OH)_2^0$ species dominate at pH = 11, the $Cu(OH)_3^-$ and $Cu(OH)_4^{2-}$ species begin to increase at pH > 11.5, as observed in the SDDs with changes in solubility (see Figures 1 and 2). The crystalline structure and surface morphologies obtained at pH > 11.5 values can be achieved by adding OH$^-$ ions through NH$_4$OH until reaching the desired pH value. More OH$^-$ ions make the $Cu(OH)_3^-$ and $Cu(OH)_4^{2-}$ (growth units) species appear in the SDDs, and major solubility changes are observed in the SDs. This excess of OH$^-$ ions produces a vertical growth of the pointed end sheets aligned to the *z*-axis on the CuO surface.

### 4.3. X-ray Photoelectron Spectroscopy (XPS)

The chemical analysis of the prepared MWCBD-CuO films and their possible ligands were analyzed using an X-ray photoelectron spectroscopy (XPS) technique. The XPS spectra of all samples were calibrated to carbon C1s at 275.8 eV. Figure 5 shows the deconvolution of the O1s core level spectra for the films prepared at different pH values. Different colors correspond to the different pH values. The dashed lines indicate the binding energy attributed to the CuO lattice oxygen (black dashed line) positioned at 529.7 eV and the CuO defective oxygen (green dashed line) positioned from 531 eV to 531.6 eV, this state was named as the non-lattice oxygen state and is related to the high density of defects within the CuO [24–27]. In Figure 5, note that the XPS spectra for films with a pH from 11.5 to 13.5 present two deconvoluted peaks corresponding to the CuO lattice oxygen and the CuO defective oxygen, respectively. On the other hand, the deconvoluted peak at 530.16 eV corresponds to the CuO film obtained at pH = 11, closer to the $Cu_2O$ positioned at 530.2 eV [28]. Other deconvoluted peaks are observed for the CuO films located at 532.9 eV (pH = 11) and 532.1 eV (pH = 11.5). These peaks can be assigned to the binding energy of residual carbon with oxygen (C-O) and oxygen in the substrate. The MWCBD-CuO films obtained present the O1s core level peaks belong to the CuO material and a high density of oxygen defects, which indicate the nanostructure nature of the films [29].

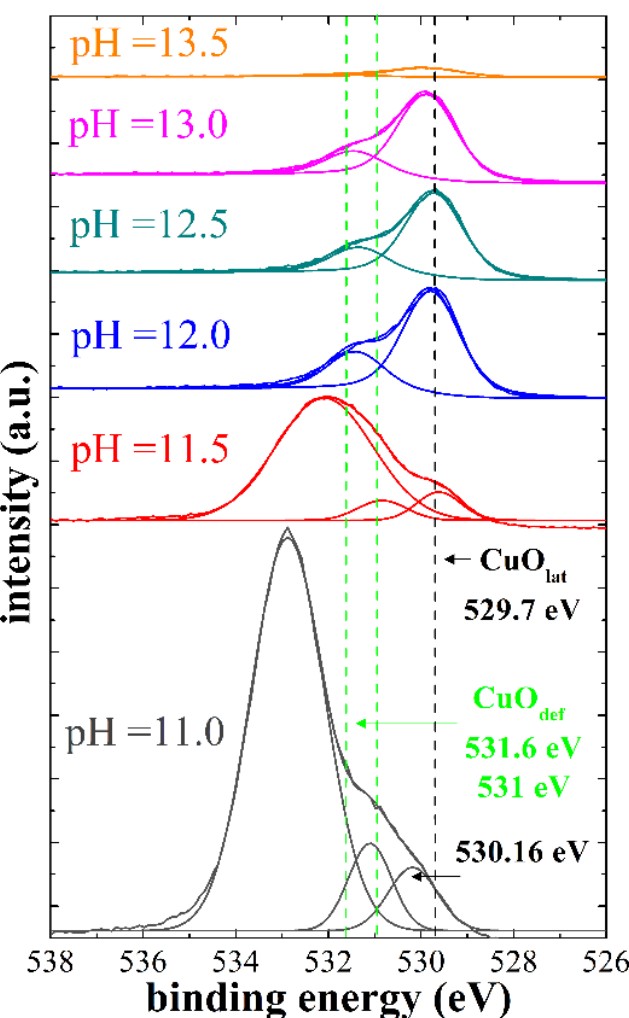

**Figure 5.** Deconvoluted peaks of the high-resolution XPS spectra of the O1s core level of the CuO films obtained at different pH values.

Figure 6 shows the XPS spectra of the Cu2p core level of the CuO films for the different pH values. The dashed lines indicate the binding energy of the $Cu2p_{3/2}$ attributed to the

CuO copper bond positioned at 933.4 eV [24–27]. All the XPS spectra in Figure 6 show the $Cu2p_{3/2}$ peak of the Cu-O ligand. In addition, a satellite peak is shown, which is distinctive to the CuO in the Cu2p core level.

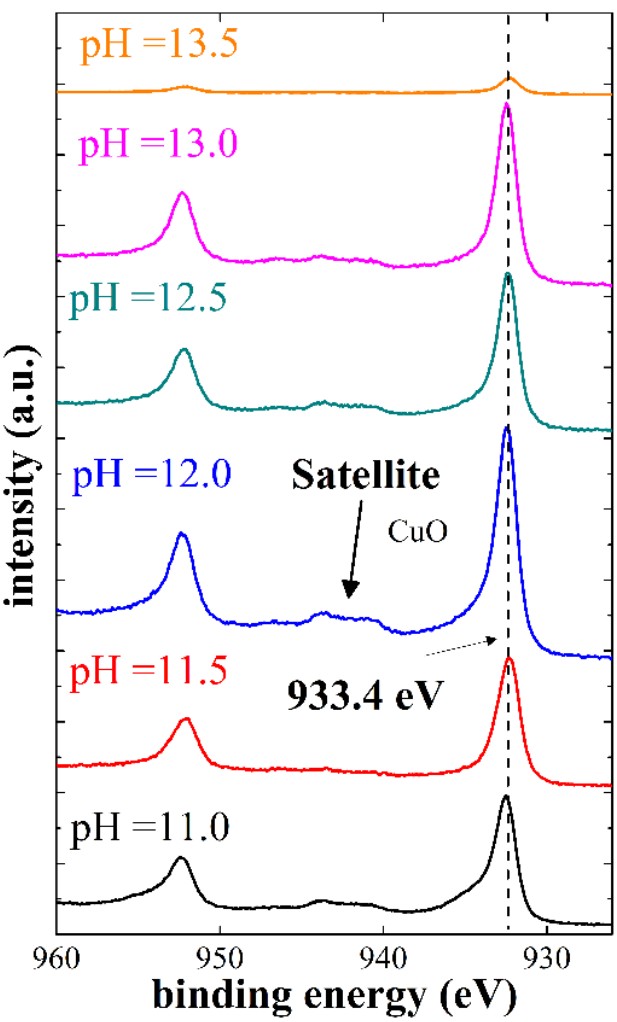

**Figure 6.** XPS spectra of Cu2p core level of the CuO films obtained at different pH values.

Table 1 shows the concentration (at%) of the deconvoluted peaks of O1s (Figure 5) and the $Cu2p_{3/2}$ (Figure 6) of the CuO films obtained at different pH values. From Table 1 we can appreciate that the MWCBD-CuO films were obtained with a Cu/O stoichiometry of 50/50 at%, except for samples at pH = 11 and pH = 13.5, which show an excess of oxygen. In addition, the O1s (Figure 5) and $Cu2p_{3/2}$ (Figure 6) peaks belonging to the samples at pH = 13.5 present a lesser intensity and lower copper and oxygen concentration. Moreover, the deconvoluted peak of the film with pH = 11 is much closer to the $Cu_2O$ oxide.

**Table 1.** Concentration (at%) of O1s and $Cu2p_{3/2}$ of the MWCBD-CuO films obtained at different pH values.

| pH Value (Concentration) | | pH = 11 (at%) | pH = 11.5 (at%) | pH = 12 (at%) | pH = 12.5 (at%) | pH = 13 (at%) | pH = 13.5 (at%) |
|---|---|---|---|---|---|---|---|
| O1s | $CuO_{lat}$ [+] | 25.65 | 37.85 | 34.82 | 39.75 | 39.58 | 47.59 |
| | $CuO_{def}$ [++] | 31.21 | 13.18 | 13.06 | 12.41 | 11.48 | 10.81 |
| Total O | | 56.86 | 51.03 | 47.88 | 52.16 | 51.06 | 58.40 |
| Total Cu | $Cu2p_{3/2}$ | 43.14 | 48.97 | 52.12 | 47.84 | 48.94 | 41.60 |
| Cu/O ratio | | 0.76 | 0.96 | 1.09 | 0.92 | 0.96 | 0.71 |

[+] lattice oxide and [++] defective oxide.

### 4.4. Optical Properties and Band-Gap Energy Determination

The bandgap energy ($E_g$) of the MWCBD-CuO films was determined through optical absorption measurements on the films. Since CuO is a semiconductor material with a direct bandgap energy, its value can be determined by the relation:

$$\alpha^2 = A(h\nu - E_g) \tag{18}$$

where $\alpha$ is the absorption coefficient, $h$ is Planck's constant, $\nu$ is the frequency of the incident light, and $A$ is a constant. The plot of $\alpha^2$ vs. $h\nu$ can be obtained and the $E_g$ value is determined by the intersection of the linear portion of the absorption curve with the energy axis ($h\nu$). The absorbance is proportional to the absorption coefficient ($\alpha \sim Abs$), so for practical purposes $\alpha^2 \sim Abs^2$. Thus, the $E_g$ value is estimated by the traced-line along the absorption edge and its intersection at $Abs^2 = 0$.

Figure 7 shows the plot of $Abs^2$ vs. $h\nu$ for the CuO films obtained at different pH values. The resulting $E_g$ values are listed in Table 2. For the films with pH = 11 and pH = 13.5, the $E_g$ value was not obtained because of their thinness and/or their poor optical quality. For the other samples, the $E_g$ values are closer to the CuO bulk value (1.2 eV), such as for the films at pH = 12 and pH = 12.5.

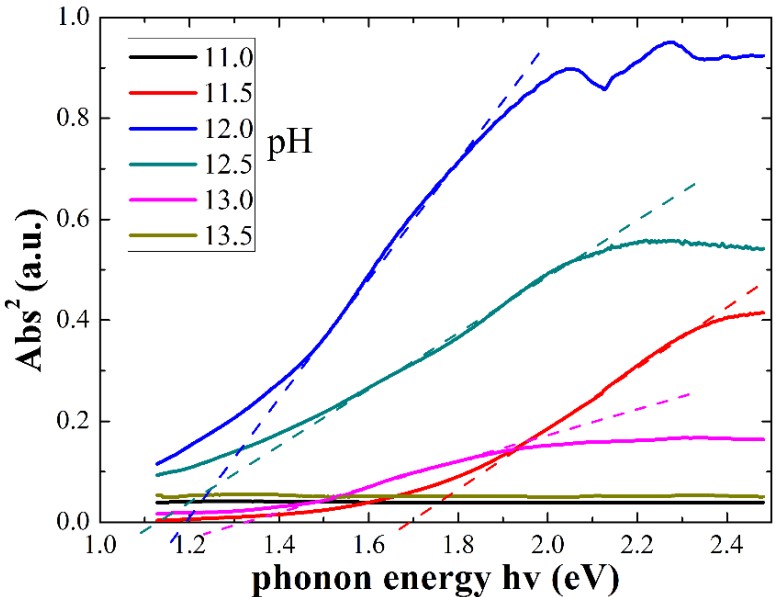

**Figure 7.** Plot of $Abs^2$ vs. $h\nu$ for obtaining the $E_g$ values of the MWCBD-CuO films obtained at different pH values.

**Table 2.** Bandgap energy ($E_g$) values of the MWCBD-CuO films deposited at different pH values.

| pH Value | 11 | 11.5 | 12 | 12.5 | 13 | 13.5 |
|---|---|---|---|---|---|---|
| $E_g$ (eV) | – | 1.68 | 1.2 | 1.16 | 1.34 | – |

## 5. Discussion

The crystalline structure of the prepared MWCBD-CuO films presents a monoclinic phase according with the CuO standard powder diffraction file. For the film prepared at pH = 11, no crystalline structure was observed, and the optical bandgap energy could not be determined due to the poor optical quality of the film. In addition, the $Cu(OH)_2^0$ species dominate in the SDDs at this pH, which is responsible for the CuO nucleation. For this reason, the film did not grow properly. By adding more $NH_4OH$ to the aqueous solution, more $OH^-$ ions are concentrated in the solution, increasing as a consequence the pH to 13, where the $Cu(OH)_3^-$ and $Cu(OH)_4^{2-}$ species appear and act as a nucleus of growth, and thus the CuO deposition rate increases, enhancing the growth of the CuO films. The XRD,

XPS, and optical results confirmed that the MWCBD-CuO films obtained at these higher pH values present better properties. For the deposited sample at pH = 13.5, the $Cu(OH)_4^{2-}$ species dominated; however, less copper and oxygen were found on the surface, as was shown by the XPS results, and the optical band-gap energy could be obtained, indicating that this CuO film was optically poor. The growing mechanism of the CuO films can be explained and understood through the $Cu(OH)_2^0$, $Cu(OH)_3^-$, and $Cu(OH)_4^{2-}$ intermediate species, where nucleation and growth happen at the same time. These conditions of the aqueous solution guarantee the adequate growth of the CuO film. On the other hand, the prepared copper oxide shows good nanostructured and optical qualities, which make it a potential absorbing material for low-cost photovoltaic applications.

## 6. Conclusions

CuO films were obtained by the microwave activated chemical bath method. The chemical components used in the solution were copper acetate and aqueous ammonia. An analytical model was developed in order to construct the species distribution diagrams (SDDs) and the solubility diagrams (SDs), by varying the pH from 11 to 13.5 by adding $NH_4OH$. From the SDDs and SDs, different chemical species in the bath were observed as a function of the pH. By using the chemical conditions observed in the SDDs and SDs, a series of films were deposited as a function of pH. The growing mechanism processes of the CuO films can be explained through the $Cu(OH)_2^0$, $Cu(OH)_3^-$, and $Cu(OH)_4^{2-}$ species, as confirmed by the experimental results. The deposited CuO films present crystalline a structure with monoclinic phase, in accordance with the CuO standard. The CuO films showed a characteristic surface morphology, such as pointed end sheets, and depending on the pH value. The chemical composition and the Cu/O stoichiometry of the films were 50/50 at%. Some of the oxygen found on the CuO films' surface presented a high density of defects, indicating the nanostructured nature in the film. The optical bandgap energy value measured for the films was similar to the CuO bulk value of 1.2 eV.

**Author Contributions:** R.G.I.: supervision, conceptualization, validation, resources, writing—review and editing, I.J.G.P.: investigation, methodology and experiments, formal analysis, visualization, data curation, writing—original draft preparation, C.T.N.: investigation, formal analysis, writing—review and editing, C.M.-R.: investigation, R.R.T.: investigation, R.S.-G.: investigation, E.R.: investigation, validation, writing—review and editing, I.G.: investigation, A.C.: investigation, G.G.-S.: investigation, I.O.A.: conceptualization, validation, writing—review and editing. All authors have read and agreed to the published version of the manuscript.

**Funding:** This research received no external funding.

**Institutional Review Board Statement:** Not Applicable.

**Informed Consent Statement:** All authors have read and agreed to the published version of the manuscript.

**Data Availability Statement:** All data are available in the manuscript.

**Acknowledgments:** This work was supported by the project 100319688-VIEP2019. The authors thank LANNBIO (Cinvestav-Mérida) for the analysis and the technical help of Wilian Cauich for the XPS measurements.

**Conflicts of Interest:** The authors declare no conflict of interest.

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
