# Peer review of "Copper Oxide Films Deposited by Microwave Assisted Alkaline Chemical Bath"

_crystals, doi:10.3390/cryst11080968_

Round 1

Reviewer 1 Report

The manuscript can be accepted for the publication as it is.

What is the main question addressed by the research?

The main problem addressed in the manuscript is the growth and the quality of copper oxide films via chemical bath deposition.  

Is it relevant and interesting?

The research is important because there is a general effort in producing elements of smart materials that can be used in devices of the next generation (e.g., magnetic, electronic and optical systems).  

How original is the topic?

The topic has been already studied by the authors but in the current manuscript they show and discuss a deposition assisted by microwaves that is potentially relevant for industrial implementations.  

What does it add to the subject area compared with other published material?

The investigation adds interesting results on the CuO film quality as a function of the chemical bath pH.  
Is the paper well written?

The paper is well written and without typos.  

Is the text clear and easy to read?

The manuscript is available even from not specialists in this research topic.
Are the conclusions consistent with the evidence and arguments presented? The conclusions are consistent and focused on the main target as discussed in the manuscript introduction.  

Do they address the main question posed?

Yes, they address the main question posed in the research work.

Author Response

Journal                        Crystals (ISSN 2073-4352)

Manuscript ID                        crystals-1337210

Answers to Reviewer 1

We are confused by the fact that you comment that the manuscript requires extensive editing of English language and style, then you suggest that the manuscript can be accepted for the publication as it is and its well written and without typos. Just in case, the manuscript was checked again including results, and the new version is attached with the response to all reviewers. 

The growth and the quality of copper oxide films depends on the pH in the aqueous solution, where nucleation and growth need to occur at the same time to guarantee quality on copper oxide films properties.  

Reviewer 2 Report

The authors report the synthesis of the copper oxide by microwave-assisted chemical bath method. With different analysis equipments, they obtained the details of the crystalline structure, surface morphology, chemical composition, and optical bandgap energy. However, some mechanisms and illustrations in figures do not seem complete. There are some comments and questions listed below.

  1. In line142, Zn(OH)2  wasnot mentioned in previous paragraphs; thus, readers might feel confused when reading the article. Also, there is a typo in Eq12, which should be Eq
  2. Lots of vague chemical formulas in Figure 1, such as, Cu2+ ,  Cu(NH3)2+, Cu(OH)+ and so on.
  3. It should be specified that which part of the surface morphology wascaused by the solubility changes and by the influence of the zinc-hydroxy complexes in lines 271-
  4. In Figure 4, the crystals from pH12 to pH13 became smaller and finer; however, the crystals at pH13.5 became bigger. Could the authors explain the phenomenon?
  5. From the SEM image in figure 4, the authors mentioned that the surface morphologies can be changed by adding OH- ions to reach the desired pH value. What is the mechanism of how OH- ions affect the surface morphologies from pH=11 to pH= 11.5?
  6. In lines 295-297, it was mentioned that the peaks located at 532.9 eV and 532.1 eV correspond to the binding energy of residual carbon with oxygen (C-O) and oxygen in the substrate. Why is the intensity of these peaks greater than that of the peak of CuO lattice oxygen? Why does the intensity of these peaks decrease as the pH value increases?
  7. Why are the O1s (Fig.5) and Cu2p3/2 (Fig. 6) peaks for the samples at pH=13.5 of lower intensity, and lower copper and oxygen concentration in lines 313-314. How did this happen?
  8. In Table 2 showing the bandgap energy (Eg) values of the MWCBD-CuO films at different pH values, why does the bandgap energy drop first and then rise at pH=11 to pH= 13?
  9. The format of Table 2 should be adjusted.
  10. In line 351, the authors mentioned that the Egvalue couldn’t be obtained due to their low thickness. I was wondering whether there is further discussion on the relationship between CuO deposition rate and thickness to pH 
  11. The conclusion says that the copper oxide has good nanostructures, optical properties and low cost advantages; in fact, it had better specify the actual applications of this experiment under different pH values.

Author Response

Journal                        Crystals (ISSN 2073-4352)

Manuscript ID                        crystals-1337210

Answers to Reviewer 2 comments

The manuscript was English language and style checked

  1. In line142, Zn(OH)2  wasnot mentioned in previous paragraphs; thus, readers might feel confused when reading the article. Also, there is a typo in Eq12, which should be Eq

Ans.    Eq. 12 and Zn(OH)2 is wrong in line 142, it was changed to Cu(OH)2 and Eq. 11

  1. Lots of vague chemical formulas in Figure 1, such as, Cu2+,  Cu(NH3)2+, Cu(OH)+and so on.

Ans.    New plots were added to Figure 1.

  1. It should be specified that which part of the surface morphology wascaused by the solubility changes and by the influence of the zinc-hydroxy complexes in lines 271-

Ans.    The paragraphs in lines 271- were put together to improve the explanation of why the changes in surface morphology wascaused by the solubility and the zinc-hydroxy complexes.

  1. In Figure 4, the crystals from pH12 to pH13 became smaller and finer; however, the crystals at pH13.5 became bigger. Could the authors explain the phenomenon?

Ans.    The crystals at pH=13.5 became bigger because in the SDDs (Fig. 1) the Cu(OH)42- specie dominate and acts as growth unit, however at this pH the CuO film have not good properties. The CuO film growth need nucleation (Cu(OH)2). The discussion section gives this explanation.

  1. From the SEM image in figure 4, the authors mentioned that the surface morphologies can be changed by adding OH-ions to reach the desired pH value. What is the mechanism of how OH- ions affect the surface morphologies from pH=11 to pH= 11.5?

Ans.    The mechanism of how OH- ions affect the surface morphologies from pH=11 to pH=11.5 is because at pH=11.5 the growth unit Cu(OH)3- increase its relative fraction (r) and nucleation (Cu(OH)2) and growth (Cu(OH)3-) happen at the same time. The discussion section gives this explanation.

  1. In lines 295-297, it was mentioned that the peaks located at 532.9 eV and 532.1 eV correspond to the binding energy of residual carbon with oxygen (C-O) and oxygen in the substrate. Why is the intensity of these peaks greater than that of the peak of CuO lattice oxygen? Why does the intensity of these peaks decrease as the pH value increases?

Ans.    The reason is because the thickness of the CuO films growth at pH=11 and pH=11.5 is lower compared with the CuO films growth at pH>11.5 and the equipment for the X-ray photoelectron spectroscopy (XPS) measurements made an erosion to the samples, which touch the substrate (glass).

  1. Why are the O1s (Fig.5) and Cu2p3/2 (Fig. 6) peaks for the samples at pH=13.5 of lower intensity, and lower copper and oxygen concentration in lines 313-314. How did this happen?

Ans.    The CuO film obtained at pH=13.5 have poor adherent, physically this film comes off on contact. However the characterization was made on this film.   

  1. In Table 2 showing the bandgap energy (Eg) values of the MWCBD-CuO films at different pH values, why does the bandgap energy drop first and then rise at pH=11 to pH= 13?

Ans.    The trend of the Eg values as pH increase, first drop and then rise, is because the optical quality obtained in the films. For pH=12 and pH=12.5 the CuO films Eg values are closer to the CuO bulk value (1.2 eV).

  1. The format of Table 2 should be adjusted.

Ans.    The format of Table 1 and 2 were adjusted.

  1. In line 351, the authors mentioned that the Egvalue couldn’t be obtained due to their low thickness. I was wondering whether there is further discussion on the relationship between CuO deposition rate and thickness to pH 

Ans.    The Eg value for the CuO film deposited at pH=11 could not be obtained because only nucleation (Cu(OH)2) happen at this pH. Nucleation is not sufficient for enhancing the growth of the CuO films i.e. the thickness.

  1. The conclusion says that the copper oxide has good nanostructures, optical properties and low cost advantages; in fact, it had better specify the actual applications of this experiment under different pH values.

Ans.    The experiment was designed to show the way to obtain copper oxide films with good properties. The experiments are reproducible, easy, low cost and scalable for industrial implementation. On the other hand, copper oxide material have many applications in electronic, magnetic and optical devices of the next generation.     

Reviewer 3 Report

I find the paper nice and well written, with a very complete characterization. I just have some minor points, namely:

  • If the study deals with testing performed with variation of pH, it is  not clear on why only pHs from 11 to 13.5 were tested. What happens at lower values?
  • Also the variation of pH should be mentioned in the title.
  • Talbe 1, XPS - Cu2p spectra allows to also determine the amount of Cu2+/Cu+ by analysis of peak areas. Moreover, the Cu2p XPS spectra (Fig 6) should be deconvolutes, just like the O1s spectra (Fig 5).
  • Also it is not clear the influence of microwave. I would like to see a comprison, for one value of pH, at least without microwave.

Author Response

Journal                        Crystals (ISSN 2073-4352)

Manuscript ID                        crystals-1337210

Answers to Reviewer 3 comments

The manuscript was English language and style checked, and a new version is attached.

  • If the study deals with testing performed with variation of pH, it is  not clear on why only pHs from 11 to 13.5 were tested. What happens at lower values?

Ans.    The copper oxide films growth via chemical bath requires an alkaline medium i.e. hydroxide ions (OH-) which provides the oxygen for the metallic oxide. For lower pH values than 11, the film will probably not deposit on the substrate.     

  • Also the variation of pH should be mentioned in the title.

Ans.    The title “Copper oxide films deposited by microwave assisted chemical bath” was changed to “Copper oxide films deposited by microwave assisted alkaline chemical bath”

  • Talbe 1, XPS - Cu2p spectra allows to also determine the amount of Cu2+/Cu+ by analysis of peak areas. Moreover, the Cu2p XPS spectra (Fig 6) should be deconvolutes, just like the O1s spectra (Fig 5).

Ans.    The XRD results shows three main peaks that match the standard patterns of the monoclinic phase of copper oxide (JCPDS No. 80-1268). Thus, the deconvolution of the Cu2p XPS spectra is not necessary. The O1s XPS spectra deconvoluted peaks (Fig. 6) found the defective oxygen in the films.

  • Also it is not clear the influence of microwave. I would like to see a comprison, for one value of pH, at least without microwave.

Ans.    The starting aqueous precursor solutions were prepared at room temperature at the desirable pH value. Then, they were put in a microwave oven at full power during 5 minutes. The copper oxide film were obtained. Microwave irradiation increase the reaction rate for several orders of magnitude, for conventional chemical bath this not happen and the copper oxide films should be show different properties.             
